**PLOS** NEGLECTED TROPICAL DISEASES

# SARS-CoV-2 seropositivity and COVID-19 among 5 years-old Amazonian children and their association with poverty and food insecurity

**Marcelo U. Ferreira**[1], **Isabel Giacomini**[2], **Priscila M. Sato**[2], **Barbara H. Lourenço**[2], **Vanessa C. Nicolete**[1], **Lewis F. Buss**[3], **Alicia Matijasevich**[4], **Marcia C. Castro**[5], **Marly A. Cardoso**[2]*, for the MINA-Brazil Working Group¶

1 Department of Parasitology, Institute of Biomedical Sciences, University of São Paulo, São Paulo, Brazil, 2 Department of Nutrition, School of Public Health, University of São Paulo, São Paulo, Brazil, 3 Institute of Tropical Medicine, University of São Paulo, São Paulo, Brazil, 4 Departamento de Medicina Preventiva, Faculdade de Medicina (FMUSP), Universidade de São Paulo, São Paulo, Brasil, 5 Department of Global Health and Population, Harvard T.H. Chan School of Public Health, Boston, Massachusetts, United States of America

¶ A full list of members of the MINA-Brazil Study Working Group is provided in the Acknowledgments section.
* marlyac@usp.br

**Data Availability Statement:** Our dataset is publicly available at the USP repository: https://repositorio.uspdigital.usp.br/handle/item/360.

## Abstract

### Background

The epidemiology of childhood SARS-CoV-2 infection and COVID-19-related illness remains little studied in high-transmission tropical settings, partly due to the less severe clinical manifestations typically developed by children and the limited availability of diagnostic tests. To address this knowledge gap, we investigate the prevalence and predictors of SARS-CoV-2 infection (either symptomatic or not) and disease in 5 years-old Amazonian children.

### Methodology/Principal findings

We retrospectively estimated SARS-CoV-2 attack rates and the proportion of infections leading to COVID-19-related illness among 660 participants in a population-based birth cohort study in the Juruá Valley, Amazonian Brazil. Children were physically examined, tested for SARS-CoV-2 IgG and IgM antibodies, and had a comprehensive health questionnaire administered during a follow-up visit at the age of 5 years carried out in January or June-July 2021. We found serological evidence of past SARS-CoV-2 infection in 297 (45.0%; 95% confidence interval [CI], 41.2–48.9%) of 660 cohort participants, but only 15 (5.1%; 95% CI, 2.9–8.2%) seropositive children had a prior medical diagnosis of COVID-19 reported by their mothers or guardians. The period prevalence of clinically apparent COVID-19, defined as the presence of specific antibodies plus one or more clinical symptoms suggestive of COVID-19 (cough, shortness of breath, and loss of taste or smell) reported by their mothers or guardians since the pandemic onset, was estimated at 7.3% (95% CI, 5.4–9.5%). Importantly, children from the poorest households and those with less educated

**Funding:** The MINA-Brazil Study has been funded by the National Council for Scientific and Technological Development of Brazil (CNPq, https://www.gov.br/cnpq/pt-br; grant number 407255/2013-3 to M.A.C.); and the São Paulo State Research Foundation (FAPESP, https://fapesp.br/en; grant number 2016/00270-6 to M.A.C.). M.U.F, A.M. and M.A.C. are recipients of CNPq senior research scholarships; I.G., P.M.S., and V.C.N. are supported by FAPESP scholarships (2021/01688-2, 2017/05651-0, and 2020/07020-0 respectively). The funders had no role in study design, data collection and interpretation, or the decision to submit the work for publication.

**Competing interests:** The authors have declared that no competing interests exist.

mothers were significantly more likely to be seropositive, after controlling for potential confounders by mixed-effects multiple Poisson regression analysis. Likewise, the period prevalence of COVID-19 was 1.8-fold (95%, CI 1.2–2.6-fold) higher among cohort participants exposed to food insecurity and 3.0-fold (95% CI, 2.8–3.5-fold) higher among those born to non-White mothers. Finally, children exposed to household and family contacts who had COVID-19 were at an increased risk of being SARS-CoV-2 seropositive and–even more markedly–of having had clinically apparent COVID-19 by the age of 5 years.

## Conclusions/Significance

Childhood SARS-CoV-2 infection and COVID-19-associated illness are substantially underdiagnosed and underreported in the Amazon. Children in the most socioeconomically vulnerable households are disproportionately affected by SARS-CoV-2 infection and disease.

### Author summary

The epidemiology of childhood COVID-19 in the tropics remains a relatively neglected research topic, in part because SARS-CoV-2 typically causes fewer severe illnesses, hospitalizations, and deaths in children than in adults. Here we show that 45% of 660 participants in a birth cohort study in the Brazilian Amazon had SARS-CoV-2 antibodies at the age of 5 years, although only 5% of them reported previously diagnosed COVID-19 episodes–implying that as many as 8 in 9 SARS-CoV-2 infections had remained undiagnosed in these young children. Only 16% of the seropositive children had reportedly experienced cough, shortness of breath, and/or loss of taste or smell. The most socioeconomically vulnerable participants were more likely to have experienced SARS-CoV-2 infection and overt COVID-19 by the age of 5 years. Importantly, children exposed to household food insecurity, which affects 54% of our study participants, had their COVID-19 risk increased by 76%.

## Introduction

Children account for a relatively small fraction of total COVID-19 cases, hospitalizations, and deaths due to SARS-CoV-2 infection worldwide [1], but this fraction has recently increased as more adults receive their COVID-19 vaccines and become partially protected from infection and disease [2]. Although severe illness may occasionally develop, children with SARS-CoV-2 infection typically remain asymptomatic or have mild symptoms [3–5]. Nevertheless, both symptomatic and asymptomatic children can carry relatively high concentrations of viral mRNA [6] and replicating SARS-CoV-2 [7], constituting a potential infectious reservoir of public health importance. Young children, who are not routinely vaccinated, appear to be more likely to transmit SARS-CoV-2 to household contacts once infected, compared with older children and adolescents [8].

The COVID-19 crisis in Brazil has been most dramatic across the Amazon, a region that covers 60% of the country's territory, where public health facilities were already operating near full capacity before the pandemic and very few intensive care unit beds were available. The emergence of the Gamma (formerly known as P.1) variant of concern in Manaus [9], the largest city in the Amazon, was followed by a dramatic upsurge in mortality across the region

during the second COVID-19 epidemic wave in early 2021 [10,11]. Official COVID-19 case notification statistics in the Amazon are imprecise due to the shortage of diagnostic molecular and antigen-detection tests, but SARS-CoV-2 antibody data can be used to estimate the proportion of mild or asymptomatic infections that remained undetected and the proportion of infections that led to overt disease, health facility visits, and hospitalization.

The epidemiology of SARS-CoV-2 infection and COVID-19 in young children remains relatively understudied [1], especially in hard-hit communities in tropical, low to middle-income countries. To address this knowledge gap, this exploratory study examines the prevalence of SARS-CoV-2 antibodies and the period prevalence of COVID-19 at the age of 5 years among participants in an ongoing population-based birth cohort study in Amazonian Brazil [12]. We show that nearly half of the children had serological evidence of prior SARS-CoV-2 infection, but very few had COVID-19 episodes previously diagnosed, suggesting that SARS-CoV-2 infections in young children are substantially underreported in this high-transmission setting.

## Materials and methods

### Ethics statement

All mothers or children's parents or guardians (if mothers were <18 years old) provided written informed consent. The study protocol was approved by the institutional review board of the School of Public Health, University of São Paulo (# 872.613, 2014; #2.358.129, 2017).

### Study design and population

The Maternal and Child Health and Nutrition in Acre, Brazil (MINA-Brazil) study is a prospective, population-based birth cohort set-up in 2015 to examine the impact of a wide range of early exposures on child growth and development in the Amazonian municipality of Cruzeiro do Sul, next to the Brazil-Peru border [12]. The study site and population are described in detail in S1 File. Fig 1 shows the study flow diagram. Briefly, mother-baby pairs were enrolled at pregnancy in public antenatal clinics, or at birth in the Women and Children's Hospital of Juruá Valley, the only maternity hospital of Cruzeiro do Sul, where 96% of all local deliveries take place [13]. At delivery, mothers were interviewed to obtain sociodemographic information [12] such as the duration of mother's schooling self-reported mother's skin color, mother's occupation, and whether the family is currently supported by the *Bolsa Família* conditional cash transfer program [14]. An assets-based wealth index [15] was used as a proxy of socioeconomic status.

Here we focus on sociodemographic and morbidity data obtained during the 5-year follow-up visit. We searched for evidence of household food insecurity, defined when household members have "no access to sufficient safe and nutritious food to meet their dietary needs and food preferences for an active and healthy life" [16], by using a validated, 5-question version of the Brazilian Food Insecurity Scale [17]. History of prior COVID-19 episodes was collected for children and their household and family contacts (parents, siblings, grandparents, or neighbors). Venous blood was collected for SARS-CoV-2 antibody testing. Information was obtained from 695 (56.0%) out of 1240 eligible children. Children participating in the 5-year assessment and those lost to follow-up (n = 514) had similar perinatal characteristics regarding sex, gestational age, preterm birth, and birth weight, but differed significantly in the proportion of children from poorest families (36.2% vs. 44.1%) and of mothers with ≤9 years of schooling (26.4% vs. 47.5%), respectively, with $P$ <0.01 ($\chi^2$ test) for both.

Follow-up assessment of 5 years-old children was carried out in two rounds. The first took place between 15 and 31 of January 2021 and targeted children born between July and December 2015; the second, between 18 of June and 4 of July 2021, targeted children born between

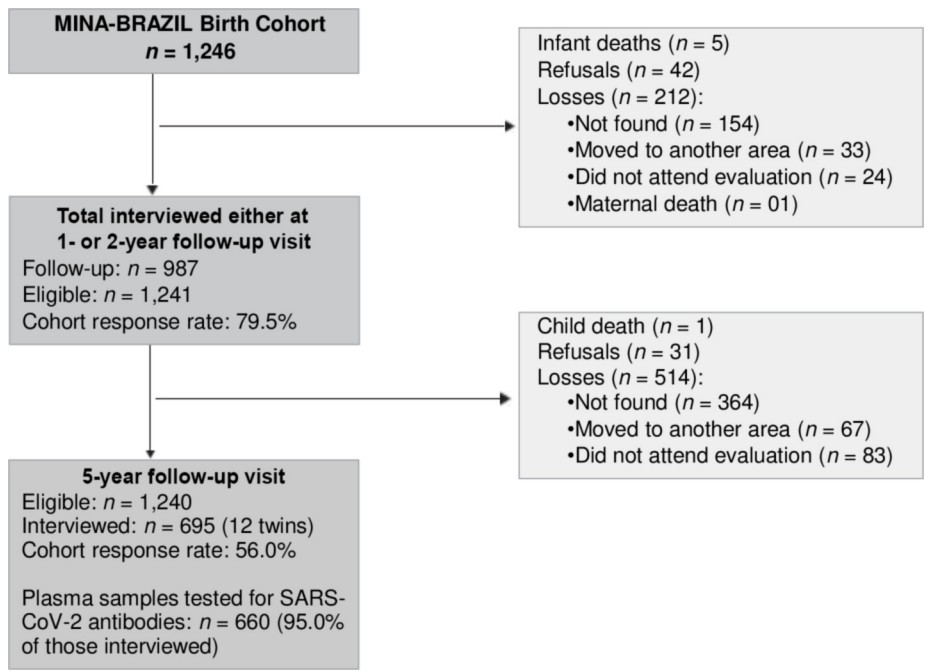

**Fig 1. Study flowchart.** Between July 2015 and June 2016, pregnant women attending antenatal clinics or admitted for delivery to the maternity ward of the Women and Children's Hospital of Juruá Valley in Cruzeiro do Sul, Brazil, were invited to participate. Reasons for exclusion and the final number of subjects analyzed for SARS-CoV-2 antibodies at the age of 5 years are indicated.

January and June 2016. At the time of the follow-up assessment, children's age ranged between 58.5 and 71.9 months (average, 63.5 months), with a significant difference between the first and the second round (63.2 vs. 63.8 months respectively, $P = 0.003$, Student's $t$ test). No study participant had been vaccinated against COVID-19 at the time of the study rounds; vaccination of children aged 5–11 years started in Brazil on 14 January, 2022. Child weight and height were measured in duplicate by trained health professionals [18] as described in detail in S1 File. Anthropometric data were processed with the Anthro software (https://www.who.int/childgrowth/en/), which uses the World Health Organization standard curves [19]. Stunting (stature-to-age: <-2 z-scores) and overweight (body mass index [BMI]-to-age: >2 z-scores) were included as covariates in our initial analyses.

## SARS-CoV-2 antibody measurement

Plasma samples from 660 study children (95.0% of those interviewed) were tested for SARS-CoV-2 antibodies; 330 of them were from the first study round and 330 from the second round. Importantly, an upsurge in COVID-19 cases was recorded in the study site between the study rounds, lasting between February and March 2021, when the Gamma variant became widespread across Juruá Valley [20].

Total antibodies (including IgG and IgM) that recognize a recombinant protein representing the nucleocapsid antigen of SARS-CoV-2 were detected in 20-μL plasma aliquots using the Elecsys anti-SARS-CoV-2 double-antigen sandwich electrochemiluminescence assay (Roche Diagnostics, Penzberg, Germany) on a Cobas 411 analyzer (Roche Diagnostics). The Elecsys assay provides a qualitative result (reactive vs. nonreactive) but with a quantitative signal cut-off index (COI) value; a COI $\geq 1.0$ is interpreted as reactive and a COI $<1.0$ is interpreted as nonreactive. Owing to its antigen sandwich format, the Elecsys assay shows good signal

stability, with little COI decay over time and rare seroreversions following a natural infection, being ideally suited for serosurveillance [21].

## Outcome definitions

Two primary outcomes were considered: (1) presence of SARS-CoV-2 antibodies in the 5-year survey, regardless of any clinical signs and symptoms, as a proxy of SARS-CoV-2 infection any-time since the COVID-19 epidemic onset; and (2) presence of SARS-CoV-2 antibodies in the 5-year survey combined with at least one new or increased clinical sign or symptom experienced since April 2020, as a proxy of prior clinically apparent COVID-19. We specifically asked mothers or guardians whether the child had presented since April 2020 (when the first COVID-19 cases were diagnosed in Juruá Valley) at least one of the following symptoms that were among the most suggestive of COVID-19 before the emergence of the Omicron variant: cough, shortness of breath, and loss of taste or smell. We define the proportion of children fulfilling the above criteria as the "period prevalence" of clinically apparent COVID-19, since children may have experienced one or more COVID-19 episodes anytime between the pandemic onset and the follow-up visit at 5 years of age. Our analysis does not consider other common signs and symptoms such as fever, diarrhea, and vomiting, which are often reported in childhood COVID-19 [22], because they are little specific and may be found in several other locally prevalent febrile illnesses.

## Data analysis

Data were collected using tablets programmed with CSPro software (https://www.census.gov/programs-surveys/international-programs.html) and transferred to STATA 15.1 (StataCorp, College Station, TX, USA) for statistical analysis.

We report raw seroprevalence rates as point estimates and 95% confidence intervals (CIs), as well as sensitivity- and specificity-adjusted prevalence estimates along with 95% highest density intervals (HDIs). To this end, we used a Bayesian framework that propagates uncertainty in the sensitivity and specificity estimates of the test [23]–reported as 97.2% (95% CI, 95.4–98.4%) and 99.8% (95% CI, 99.3–100%), respectively [24].

Separate multiple Poisson regression models were built to identify predictors associated with each of two binary outcomes: (1) SARS-CoV-2 infection (using seropositivity as a proxy) regardless of any clinical signs and symptoms, and (2) clinically apparent COVID-19 upon serologically documented SARS-CoV-2 infection. Because children are nested into two study rounds (grouping variable "round"), which introduces dependency among observations, for each outcome we built mixed-effects Poisson regression models with random effects at the "round" level and robust variance. Variables associated with the outcome at a significance level <20% in unadjusted analysis were entered in multiple Poisson regression models. We next used a hierarchical approach based on conceptual frameworks [25] to select covariates that were retained in the final adjusted models (S1 File). Participants with missing values in categorical covariates were maintained in the model by creating a new missing-value category. Statistical significance was defined at the 5% level. Prevalence ratio (PR) estimates are provided along with 95% CIs to quantify the influence of each predictor on the outcome while controlling for all other covariates.

## Results

### High SARS-CoV-2 seroprevalence but low COVID-19-related morbidity in Amazonian children

Overall, 297 (45.0%; 95% CI 41.2–48.9%) of the 660 study participants had serological evidence of prior SARS-CoV-2 infection (Fig 2B). Not surprisingly, given the dramatic increase in

**A**

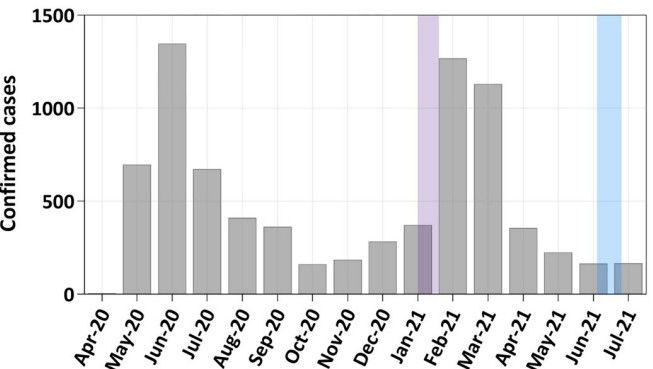

**B**

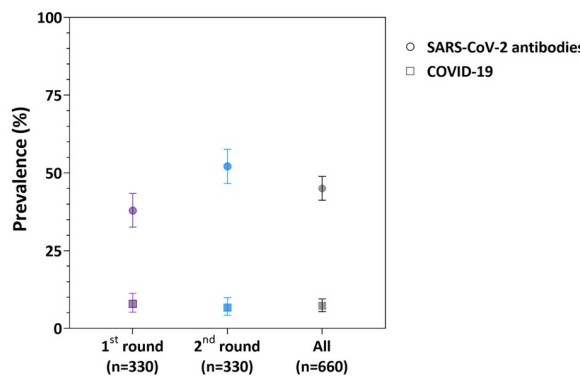

**Fig 2. COVID-19 cases in Cruzeiro do Sul, Brazil, between April 2020 and July 2021 and period prevalence of SARS-CoV-2 infection and clinically apparent COVID-19 in 5 years-old children as measured in January 2021 (first study round) and June-July 2021 (second study round).** (A) Monthly cases of COVID-19 notified in the municipality of Cruzeiro do Sul, between April 2020 and July 2021. Light purple and light blue shaded areas represent the dates of follow-up assessment in January 2021 and June-July 2021, respectively. Data source: State Secretary of Health, Acre. Data available daily at: http://saude.acre.gov.br/. (B) Circles show the proportions (%) of children positive for anti-SARS-CoV-2 antibodies in the first study round (light purple), in the second round (light blue), and in both rounds combined (grey). Squares show the period prevalence (%) of clinically apparent COVID-19 (see the main text for definition) among these same children, as estimated in the first study round (light purple), in the second round (light blue), and in both rounds combined (grey). A total of 330 children were assessed during each study round. Error bars indicate 95% confidence intervals.

COVID-19 incidence in the study site in February-March 2021 (Fig 2A), antibody positivity rates were significantly higher during the second study round (June-July 2021: 52.1%, 95% CI, 46.6–57.6%), which was carried out after the case upsurge (June-July 2021), when compared with the first round (January 2021: 37.9%, 95% CI, 32.6–43.4%), with $P < 0.01$ by the $\chi^2$ test. The sensitivity- and specificity-adjusted seroprevalence estimates were 53.6% (95% HDI, 48.0–59.2%) for June-July 2021 and 38.9% (95% HDI, 33.5–44.4%) for January 2021. Only 15 (5.1%; 95% CI, 2.9–8.2%) seropositive children had a prior medical diagnosis of COVID-19 reported by their mothers or guardians at the time of the interview. Previous laboratory confirmation of SARS-CoV-2 infection was reported for only 11 (3.7%; 95% CI, 1.9–6.5%) seropositive children, 8 by antigen-based rapid diagnostic test, and 3 by reverse-transcriptase (RT)-PCR.

COVID-19-related morbidity appeared to be infrequent in this population. Only 48 (16.2%; 95% CI, 12.2–20.8%) of the 297 seropositive children had cough, shortness of breath, and/or loss of taste or smell, according to their mothers or guardians. The relative frequencies of reported symptoms were: cough (11.4%; 95% CI, 8.1–15.6%), loss of taste or smell (3.7%; 95% CI, 1.8–6.5%), and shortness of breath (2.7%; 95% CI, 1.2–5.2%). S2 Fig shows the extent of overlap of symptoms experienced by SARS-CoV-2 seropositive children. Among the 363 SARS-CoV-2 seronegative children, only 19 (5.2%; 95% CI, 3.2–8.1%) had a report of cough, shortness of breath, and/or loss of taste or smell during the study period.

The period prevalence of clinically apparent COVID-19, defined as the presence of specific antibodies and one or more of the symptoms listed above experienced since the pandemic onset, was estimated at 7.3% (95% CI, 5.4–9.5%) by the age of 5 years. COVID-19 period prevalence estimates were similar between study rounds: 7.9% (95% CI, 5.2–11.3%) in January 2021, and 6.7% (95% CI, 4.2–9.9%) in June-July 2021 (Fig 2B).

## Higher risk of SARS-CoV-2 infection and COVID-19 among socioeconomically vulnerable children

Children living under conditions of socioeconomic vulnerability–those from the poorest households, exposed to food insecurity, and with non-White and less educated mothers–were significantly more likely to be seropositive and/or to have experienced COVID-19-related morbidity. Results of the unadjusted analysis are shown in Table 1. Children from the poorest households and those with less-educated mothers remained significantly more likely to be seropositive at the age of 5 years after controlling for potential confounders (Table 2). Similarly, children from households experiencing food insecurity and those born to non-White mothers were more likely to have had COVID-19-related morbidity by the age of 5 years, after adjustment for confounders (Table 2). Importantly, the presence of these same clinical symptoms (cough, shortness of breath, and/or loss of taste or smell) was not significantly associated with household food insecurity or mother's skin color among SARS-CoV-2 seronegative children (n = 363; S1 Table). We found evidence that the socioeconomic status (children in the intermediate, but not the highest tertile, compared with the first tertile of wealth index) and >12 years of mother's schooling were predictors of reduced risk of COVID-19-unrelated respiratory symptoms, with no clear dose-response relationship (S1 Table).

## Familial and household aggregation of SARS-CoV-2 infection and COVID-19

Consistent with substantial household-level SARS-CoV-2 transmission [8,26], children whose household and family contacts (parents, other close relatives, or neighbors) reported one or more prior COVID-19 episodes were significantly more likely to be seropositive and have experienced COVID-19 morbidity by the age of 5 years (Fig 3). Separate models were built for each child's contact (mother, father, siblings, grandparents, etc.). Positive associations were stronger for child contacts who usually live inside the child's household, such as parents and siblings. Nevertheless, significant associations were also found for grandparents and more distant relatives, as well as neighbors, who usually do not share the household with the study children (Fig 3).

## Discussion

The present cross-sectional serosurveys in the Amazon Basin of Brazil provide evidence for a substantial underreporting of SARS-CoV-2 infection among young children from this hard-

**Table 1. Prevalence (%) of SARS-CoV-2 antibodies and prevalence ratio (95% confidence interval) of clinically apparent COVID-19 in 5 years-old Amazonian children according to sociodemographic, neonatal, nutritional, and morbidity history variables.**

| Variable | Total | SARS-CoV-2 antibodies | | Clinically apparent COVID-19 | |
|---|---|---|---|---|---|
| | | Prevalence (%) | Prevalence ratio (95% CI)* | Prevalence (%) | Prevalence ratio (95% CI)* |
| Household wealth index | | | | | |
| 1 (poorest) | 222 | 52.3 | 1 | 8.6 | 1 |
| 2 | 219 | 47.5 | 0.90 (0.65–1.23) | 7.3 | 0.85 (0.63–1.16) |
| 3 (wealthiest) | 219 | 35.2 | 0.67 (0.54–0.84) | 6.0 | 0.69 (0.31–1.56) |
| | | | $P < 0.001$ | | $P = 0.369$ |
| Support from the *Bolsa Família* program | | | | | |
| No | 360 | 40.8 | 1 | 5.3 | 1 |
| Yes | 300 | 50.0 | 1.23 (1.10–1.38) | 9.7 | 1.83 (0.66–5.08) |
| | | | $P < 0.001$ | | $P = 0.245$ |
| Household food insecurity | | | | | |
| No | 306 | 43.8 | 1 | 4.9 | 1 |
| Yes | 354 | 46.1 | 1.06 (0.94–1.19) | 9.3 | 1.90 (1.18–3.06) |
| | | | $P = 0.334$ | | $P = 0.008$ |
| Mother's schooling | | | | | |
| ≤ 9 years | 172 | 48.3 | 1 | 9.9 | 1 |
| 10 to 12 years | 327 | 49.5 | 1.05 (0.95–1.15) | 6.7 | 0.68 (0.31–1.47) |
| >12 years | 146 | 32.2 | 0.68 (0.67–0.69) | 5.5 | 0.55 (0.30–1.02) |
| | | | $P < 0.001$ | | $P = 0.141$ |
| Mother's skin color | | | | | |
| White | 79 | 30.4 | 1 | 2.5 | 1 |
| Non-white | 567 | 47.3 | 1.59 (0.99–2.53) | 7.9 | 3.13 (2.67–3.68) |
| | | | $P = 0.053$ | | $P < 0.001$ |
| Mother's occupation | | | | | |
| No paid job | 409 | 48.7 | 1 | 8.1 | 1 |
| Paid job | 237 | 39.2 | 0.82 (0.79–0.86) | 5.9 | 0.73 (0.62–0.86) |
| | | | $P < 0.001$ | | $P < 0.001$ |
| Child's age (months) | 660 | – | 0.97 (0.96–0.99)** | – | 0.91 (0.91–0.92)** |
| | | | $P < 0.001$ | | $P < 0.001$ |
| Child's sex | | | | | |
| Female | 326 | 44.2 | 1 | 6.8 | 1 |
| Male | 334 | 45.8 | 1.03 (0.95–1.12) | 7.8 | 1.15 (0.59–2.27) |
| | | | $P = 0.411$ | | $P = 0.680$ |
| Birth weight (kg) | | | | | |
| 2.5 – 4.0 | 569 | 45.2 | 1 | 7.4 | 1 |
| < 2.5 | 48 | 50.0 | 1.14 (0.73–1.78) | 10.4 | 1.41 (1.12–1.77) |
| > 4.0 | 42 | 38.1 | 0.84 (0.73–0.96) | 2.4 | 0.32 (0.40–2.58) |
| | | | $P = 0.069$ | | $P = 0.266$ |
| Prematurity | | | | | |
| No | 601 | 45.6 | 1 | 7.2 | 1 |
| Yes | 59 | 39.0 | 0.85 (0.78–0.93) | 8.5 | 1.18 (0.88–1.59) |
| | | | $P = 0.001$ | | $P = 0.255$ |
| Exclusive breastfeeding | | | | | |
| ≥ 90 days | 389 | 4.0 | 1 | 8.2 | 1 |
| No | 214 | 46.7 | 1.02 (0.89–1.18) | 6.5 | 0.80 (0.48–1.32) |
| Yes | | | $P = 0.738$ | | $P = 0.373$ |
| Total breastfeeding ≥12 months | | | | | |
| No | 219 | 42.0 | 1 | 9.6 | 1 |
| Yes | 429 | 45.2 | 1.06 (0.83–1.35) | 5.6 | 0.58 (0.33–1.03) |
| | | | $P = 0.626$ | | $P = 0.063$ |
| Total breastfeeding ≥12 months | | | | | |
| No | 219 | 42.0 | 1 | 9.6 | 1 |
| Yes | 429 | 45.2 | 1.06 (0.83–1.35) | 5.6 | 0.58 (0.33–1.03) |
| | | | $P = 0.626$ | | $P = 0.063$ |

*(Continued)*

**Table 1.** (Continued)

| | | SARS-CoV-2 antibodies | | Clinically apparent COVID-19 | |
|---|---|---|---|---|---|
| Stunting at 5 years of age | | | | | |
| No | 645 | 44.8 | 1 | 7.3 | 1 |
| Yes | 15 | 53.3 | 1.20 (0.87–1.65) | 6.7 | 0.91 (0.09–9.15) |
| | | | P = 0.259 | | P = 0.940 |
| Overweight at 5 years of age | | | | | |
| No | 571 | 44.8 | 1 | 7.2 | 1 |
| Yes | 89 | 46.1 | 1.01 (0.57–1.82) | 7.9 | 1.10 (0.22–5.43) |
| | | | P = 0.963 | | P = 0.911 |
| Malaria infection within the past 12 months | | | | | |
| No | 643 | 44.8 | 1 | 7.2 | 1 |
| Yes | 17 | 52.9 | 1.14 (0.90–1.46) | 11.8 | 1.64 (1.08–2.51) |
| | | | P = 0.283 | | P = 0.021 |
| Pneumonia within the past 12 months | | | | | |
| No | 655 | 45.0 | 1 | 7.3 | - |
| Yes | 5 | 40.0 | 0.87 (0.44–1.73) | 0 | |
| | | | P = 0.687 | | |
| Anemia at 5 years of age | | | | | |
| No | 624 | 44.4 | 1 | 7.4 | 1 |
| Yes | 34 | 55.9 | 1.25 (1.12–1.39) | 5.9 | 0.80 (0.11–5.87) |
| | | | P < 0.001 | | P = 0.825 |
| Immunization status[#] | | | | | |
| Complete | 513 | 44.6 | 1 | 7.4 | 1 |
| Incomplete | 147 | 46.3 | 1.03 (0.95–1.12) | 6.8 | 0.92 (0.65–1.29) |
| | | | P = 0.417 | | P = 0.624 |

Clinically apparent COVID-19 was defined in children with SARS-CoV-2 antibodies who reported at least one of the following symptoms/signs: cough, shortness of breath, and loss of taste or smell. Totals differ due to missing values.

*95% CI = 95% confidence interval.

**PR variation per month.

[#] Immunization status was considered complete when the participant received all doses of the recommended vaccines up to 5 years of age.

hit region, with clear public health implications. For example, the misperception that young children are less susceptible to infection has been identified as a major cause of parental vaccine hesitancy [27,28], defined as parents' delay in acceptance or refusal of vaccines despite their availability [29]. In our study, 45% of the participants had SARS-CoV-2 antibodies at the age of 5 years but only 5% of them had a prior episode of COVID-19 reported by their mothers or guardians, suggesting that 8 in 9 infections remained undiagnosed–and, therefore, were not notified. As expected [3–5], most childhood infections were subclinical and only 16% of the seropositive children had reportedly experienced cough, shortness of breath, and/or loss of taste or smell since the pandemic started.

Importantly, children with household or family contacts who reported prior COVID-19 were substantially more likely to be infected and to present COVID-19-related illness (Fig 3). The positive association between COVID-19 risk in children and past infection in their mothers was particularly strong. We acknowledge that our cross-sectional analysis does not allow us to infer the direction of SARS-CoV-2 transmission between children and their close contacts. However, literature data suggest that adults are more likely than children to acquire COVID-19 from a household or family index case [26]. We hypothesize that asymptomatic young children [8], especially those with relatively high viral loads [6,7], may be a significant source of household transmission to adults in settings characterized by poor housing conditions, little compliance with social distancing, and limited availability of diagnostic testing to identify subclinical SARS-CoV-2 carriage. Whether this applies to our study participants remains to be

**Table 2. Adjusted prevalence ratios (PR) and 95% confidence intervals (95% CI) for predictors of SARS-CoV-2 infection and clinically apparent COVID-19 in 5-year-old Amazonian children, as estimated by mixed-effects multiple Poisson regression models (n = 660).**

| Variables | SARS-CoV-2 infection | | | COVID-19 | | |
|---|---|---|---|---|---|---|
| | PR | 95% CI | *P* | PR | 95% CI | *P* |
| Child's age (months) | 0.97 | 0.96–0.98 | <0.001 | 0.90 | 0.90–0.92 | <0.001 |
| Mother's skin color* | | | | | | |
| White | 1 | | | 1 | | |
| Non-white | 1.54 | 0.93–2.54 | 0.091 | 3.05 | 2.67–3.49 | <0.001 |
| Household wealth index | | | | | | |
| 1 (poorest) | 1 | - | - | - | - | - |
| 2 | 0.92 | 0.72–1.18 | 0.496 | | | |
| 3 (wealthiest) | 0.74 | 0.57–0.97 | 0.029 | | | |
| *P* for trend | | | 0.045 | | | |
| Mother's schooling* | | | | | | |
| ≤9 years | 1 | - | - | 1 | - | - |
| 10 to 12 years | 1.09 | 0.98–1.20 | 0.126 | 0.68 | 0.32–1.46 | 0.324 |
| >12 years | 0.81 | 0.66–0.99 | 0.038 | 0.66 | 0.42–1.06 | 0.086 |
| *P* for trend | | | 0.012 | | | 0.205 |
| Household food insecurity | | | | | | |
| No | - | | | 1 | - | - |
| Yes | | | | 1.76 | 1.17–2.65 | 0.007 |

Clinically apparent COVID-19 was defined in children with SARS-CoV-2 antibodies who reported at least one of the following symptoms/signs: cough, shortness of breath, and loss of taste or smell.

*Missing values: mother's skin color, n = 14; mother's schooling, n = 15.

investigated by considering (currently unavailable) information on time of infection of each household member. Nursery school attendance does not account for the high SARS-CoV-2 prevalence in our population, since <1% of the children have reportedly attended childcare facilities since the school closures in early April 2020.

The COVID-19 pandemic has exacerbated pre-existing socioeconomic and health disparities worldwide [30] which, in turn, increase the risk of SARS-CoV-2 infection and COVID-19 morbidity in children. This is due to poverty-related comorbidities (e.g., malnutrition), reduced access to healthcare, low-quality and overcrowded housing, and parents' exposure to high-risk occupations [31]. We note, however, that some SARS-CoV-2 seropositive children may have been misclassified as having experienced clinically apparent COVID-19 in our study if they had other poverty-related respiratory infections leading to cough, shortness of breath, or loss of taste or smell. Importantly, socially vulnerable people are less likely to be tested for SARS-CoV-2 infection, but more likely to be positive once tested [32]. Here we present further evidence that, as in the United States and United Kingdom [33], the socioeconomic and ethnic status are associated with SARS-CoV-2 infection and COVID-19 risk in Brazil [34,35]. The most vulnerable Amazonian children have higher SARS-CoV-2 seropositivity rates and are more likely to have experienced COVID-19-related morbidity by the age of 5 years (Table 2). Notably, study children from the poorest families are at increased risk of many other poverty-related childhood infections in addition to COVID-19 –including malaria, dengue, and intestinal and respiratory infections by other viruses, bacteria, and parasites–that may also severely affect their growth and development.

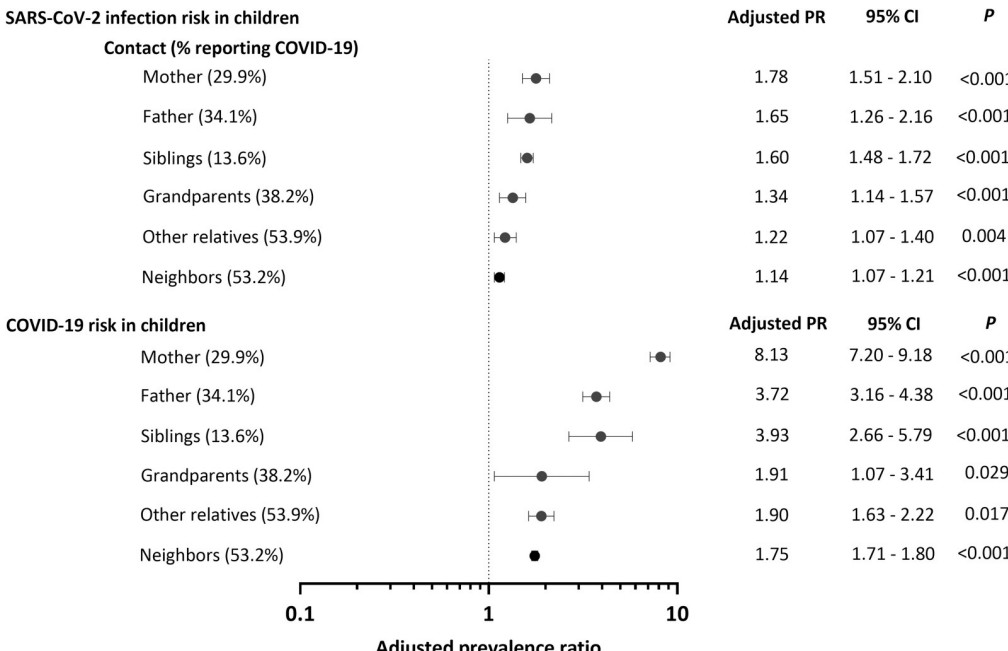

**Fig 3. Association between self-reported prior COVID-19 in household and family contacts (close relatives or neighbors) and the risk of SARS-CoV-2 infection and clinically apparent COVID-19 in 5 years-old Amazonian children.** Adjusted prevalence ratios (PR) indicate the magnitude of association between exposure to COVID-19-reporting household or family contacts and two outcomes, SARS-CoV-2 seropositivity and COVID-19, among MINA-Brazil study participants (n = 660), after controlling for the following potential confounders: child's age, mother's self-reported skin color and schooling, and household wealth index tercile for SARS-CoV-2 seropositivity; and child's age, mother's self-reported skin color and schooling, and presence of household food insecurity for clinically manifest COVID-19. Note that the denominators (numbers of children at risk) are the same in analyses with different outcomes (SARS-CoV-2 infection in the upper panel and clinically apparent COVID-19 in the lower panel). PR estimates and their respective 95% confidence intervals (95% CIs) and *P* values were derived from separate multiple mixed-effects Poisson regression models which each child contact type tested at once as an explanatory variable.

Children exposed to household food insecurity, which affects 54% of our study participants (Table 1), had their COVID-19 risk increased by 76% (Table 2). We note that food insecurity rates have increased worldwide since the onset of the COVID-19 pandemic [36,37], most markedly in low- and middle-income countries [38,39]. Compared with 2019, approximately 14 million more people in Latin America and the Caribbean were affected by hunger in 2020 (https://data.unicef.org/resources/sofi-2021/). Over one-third of households in Brazil already experienced some degree of food insecurity before the COVID-19 pandemic [40] and this proportion is estimated to have increased to 55% in 2020 (http://olheparaafome.com.br/VIGISAN_Inseguranca_alimentar.pdf), due to the reduced economic activity and increased food prices during the pandemic [38–41]. Importantly, the economic crisis aggravated by the pandemic has led families to rely on cheaper foods, most of which ultra-processed products that are rich in sugar, sodium, and fat but lack essential nutrients [40]. Strategies to mitigate poverty and food insecurity are urgently needed in the context of the current COVID-19 pandemic [36,40].

The present study has some limitations. First, only 53% of the original MINA cohort participants enrolled at birth (n = 1246) and 56% of those alive at the age of 5 years (n = 1240) had SARS-CoV-2 antibodies measured (Fig 1). Children lost to follow-up differ from those participating in the present serosurvey in key correlates of socioeconomic vulnerability, such as socioeconomic status and mother's education, suggesting that SARS-CoV-2 infection and

disease could be even more common in the original study population. Second, antibody measurements were carried out during two separate study rounds, with substantial SARS-CoV-2 transmission occurring between them (Fig 2A). Moreover, the time elapsed between potential exposure and detection of seroconversion varies widely among study participants and seroprevalence rates are not directly comparable between surveys. We used two approaches to cope with this limitation. We used a serological test with stable signal over time and rare seroreversions following a natural infection [21] to minimize the risk of differential misclassification (participants infected during the first pandemic wave might be more likely to yield a false-negative serology due to waning antibody responses). In addition, we used mixed-effects regression models to account for the clustering of participants within study rounds, as participants in each round are expected to differ in their previous exposure to (and time elapsed since) SARS-CoV-2 infection. Third, the symptoms investigated retrospectively are not necessarily specific for childhood COVID-19 and may be affected by recall bias. Some symptoms may have been caused by other locally prevalent infections experienced by their children since the COVID-19 pandemic started, such as malaria. Only the unadjusted analysis showed that the 17 children with self-reported malaria within the past 12 months were more likely to report one or more clinical signs and symptoms associated with COVID-19 (Table 1), suggesting that our definition of clinically apparent COVID-19 may have been, at least in part, confounded by malaria-associated symptoms. We acknowledge respiratory symptoms caused by diseases other than COVID-19, but also associated with poverty (S1 Table), as a potential source of residual confounding in the association between clinically apparent COVID-19 and socioeconomic status. However, only 5.2% of the SARS-CoV-2 seronegative had COVID-19-like symptoms (cough, shortness of breath, and/or loss of taste or smell) during the study period and their prevalence was not significantly associated with mother′s skin color of household food insecurity (S1 Table). Fourth, to minimize the risk of false positives, we excluded from our symptom/sign list other clinical manifestations that, although not specific, are commonly reported in childhood COVID-19. These include fever, vomiting, diarrhea, and headache [22]. For example, seropositive children with reported fever but without cough, shortness of breath, and loss of taste or smell have not been classified as having experienced clinically apparent COVID-19. Moreover, symptom reports are prone to recall bias and mild clinical manifestations may have been overlooked by mothers and guardians. Consequently, we surely have missed some clinically apparent COVID-19 cases in our study population.

## Conclusion

The present study provides new insights into the epidemiology of SARS-CoV-2 infection and COVID-19 and its association with social inequalities in young Amazonian children. Our results indicate that SARS-CoV-2 infections are frequent but substantially underreported among 5 years-old children in the Brazilian Amazon and possibly in other similar high-prevalence tropical settings. Children in the most socioeconomically vulnerable households are disproportionately affected by SARS-CoV-2 infection. Importantly, children living in households experiencing food insecurity and born to non-White mothers are more likely to have COVID-19-related morbidity once infected, further contributing to socially determined health disparities in the Amazon.

## Supporting information

**S1 Strobe Checklist. STROBE checklist.**
(DOCX)

**S1 Fig. Study site.** The map shows Brazil and Peru in South America (panel A) and the location of the municipality of Cruzeiro do Sul (dark green) in the Upper Juruá Valley region, Acre State (light green), northwestern Brazil (panel B). The urban area of Cruzeiro do Sul is shown in greater detail in panel C. Other cities and towns in the region (Mâncio Lima, Guajará, and Rodrigues Alves) are also indicated by triangles. Roads and streets are represented in light gray. Rivers are represented in blue. Figure created with QGIS software version 3.14, an open-source Geographic Information System (GIS) licensed under the GNU General Public License (https://bit.ly/2BSPB2F). Publicly available shapefiles were obtained from the Brazilian Institute of Geography and Statistics (IBGE) website (https://bit.ly/34gMq0S). Roads, streets, and rivers were obtained from the Open Street Map Foundation website (https://bit.ly/3pzh4xp). All utilized geographical data are under the Creative Commons Attribution License (CC BY 4.0). Reproduced from: Pincelli A, Cardoso MA, Malta MB, Johansen IC, Corder RM, Nicolete VC, et al. Low-level *Plasmodium vivax* exposure, maternal antibodies, and anemia in early childhood: Population-based birth cohort study in Amazonian Brazil. PLoS Negl Trop Dis. 2021;15:e0009568. doi: 10.1371/journal.pntd.0009568.
(PDF)

**S2 Fig. Venn diagram of the four considered symptoms experienced by SARS-CoV-2 seropositive children since the pandemic onset, as reported by mothers and guardians.**
(PDF)

**S1 File. Supplementary methods.**
(DOCX)

**S1 Table. Crude prevalence ratios (PR) and 95% confidence intervals (95% CI) for predictors of respiratory symptoms suggestive of respiratory infection (cough, shortness of breath, and/or loss of taste or smell in SARS-CoV-2-seronegative 5-year-old Amazonian children, as estimated by mixed-effects multiple Poisson regression models (n = 363).**
(DOCX)

## Acknowledgments

We thank all women and children who have taken part in the MINA-Brazil Study and health professionals at the Maternity Hospital, Municipal Health Secretariat, and primary health care units of Cruzeiro do Sul. Members of MINA-Brazil Working Group: Marly Augusto Cardoso (PI), Alicia Matijasevich, Bárbara Hatzlhoffer Lourenço, Jenny Abanto, Maíra Barreto Malta, Marcelo Urbano Ferreira, Paulo Augusto Ribeiro Neves (University of São Paulo, São Paulo, Brazil); Ana Alice Damasceno, Bruno Pereira da Silva, Rodrigo Medeiros de Souza (Federal University of Acre, Cruzeiro do Sul, Brazil); Simone Ladeia-Andrade (Oswaldo Cruz Institute, Fiocruz, Rio de Janeiro, Brazil), Marcia Caldas de Castro (Harvard T.H. Chan School of Public Health, Boston, USA).

## Author Contributions

**Conceptualization:** Marcelo U. Ferreira, Marcia C. Castro, Marly A. Cardoso.

**Data curation:** Marly A. Cardoso.

**Formal analysis:** Marcelo U. Ferreira, Vanessa C. Nicolete, Lewis F. Buss, Marly A. Cardoso.

**Funding acquisition:** Marly A. Cardoso.

**Investigation:** Marcelo U. Ferreira, Isabel Giacomini, Priscila M. Sato, Barbara H. Lourenço, Marly A. Cardoso.

**Methodology:** Marcelo U. Ferreira, Isabel Giacomini, Priscila M. Sato, Barbara H. Lourenço, Vanessa C. Nicolete, Lewis F. Buss, Alicia Matijasevich, Marcia C. Castro, Marly A. Cardoso.

**Resources:** Isabel Giacomini, Priscila M. Sato, Barbara H. Lourenço, Alicia Matijasevich.

**Supervision:** Marly A. Cardoso.

**Validation:** Marcelo U. Ferreira, Marly A. Cardoso.

**Visualization:** Marcelo U. Ferreira, Marly A. Cardoso.

**Writing – original draft:** Marcelo U. Ferreira, Marly A. Cardoso.

**Writing – review & editing:** Marcelo U. Ferreira, Isabel Giacomini, Priscila M. Sato, Barbara H. Lourenço, Vanessa C. Nicolete, Lewis F. Buss, Alicia Matijasevich, Marcia C. Castro, Marly A. Cardoso.

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
