## [Decision Letter · Decision Letter 0]

29 Mar 2022

Dear Dr. Cardoso,

Thank you very much for submitting your manuscript "SARS-CoV-2 infection and COVID-19 among 5 years-old Amazonian children and their association with poverty and food insecurity" for consideration at PLOS Neglected Tropical Diseases. As with all papers reviewed by the journal, your manuscript was reviewed by members of the editorial board and by several independent reviewers. In light of the reviews (below this email), we would like to invite the resubmission of a significantly-revised version that takes into account the reviewers' comments. 

Both Reviewers brought up important comments about the experimental design, the analysis and interpretation of data, and the significance of results. These points must be addressed. One reviewer had excellent suggestions for re-organizing the manuscript, particularly to ensure the manuscript is focused on very well-defined research objectives, eliminating or moving less-important information, and making the statistical and analytical approaches very clear.

We cannot make any decision about publication until we have seen the revised manuscript and your response to the reviewers' comments. Your revised manuscript is also likely to be sent to reviewers for further evaluation.

Sincerely,

Georgios Pappas

Associate Editor

Jeremy V. Camp, PhD

Deputy Editor

Reviewer comments about design and results' significance need to be addressed.

Reviewer's Responses to Questions

**Key Review Criteria Required for Acceptance?**

**Methods**

-Are the objectives of the study clearly articulated with a clear testable hypothesis stated?

-Is the study design appropriate to address the stated objectives?

-Is the population clearly described and appropriate for the hypothesis being tested?

-Is the sample size sufficient to ensure adequate power to address the hypothesis being tested?

-Were correct statistical analysis used to support conclusions?

-Are there concerns about ethical or regulatory requirements being met?

Reviewer #1: Overall, it appears authors have an ongoing birth cohort from which they are using to assess the SARS-CoV-2 antibodies in children in Amazonian Brazil. This type of study is needed to fill the current gaps in the COVID literature, but I do recommend some revisions before full manuscript acceptance. 

-Currently as the manuscript stands there is a lot of information about the birth cohort and variable acquisition that may potentially benefit from paraphrasing and including these methods in a supplementary file for individuals who are more interested in the cohort conception. Authors should primarily focus on the methods specific to this analysis of prevalence of SARS-CoV-2 antibodies among children from Brazil, this way it is more clear and concise for readers.

-Data Analysis: Authors decided to use poisson regression to identify factors associated with: 1 SARS-CoV-2 infection and 2 COVID-19 with clinical symptoms and serological positive test. Authors state that they created a separate category for missingness within covariates. This type of approach is not ideal, I suggest authors revise their analysis to include multiple imputation of the missing data for the regression models. 

- Data Analysis: Authors use an alpha level of <0.20 to select variables to adjust for in the model. This approach is typically not ideal and leads to residual confounding. Ideally authors should use another methods such as change in estimate or directed acyclic graphs, unless authors are trying to truly identify "predictors". This is not clear in the methods section of the true goal of the multivariable model. Throughout the results it appears authors were trying to build more of a causal model. This lack of clarity has made my enthusiasm for the paper decrease. 

-Laboratory methods for serological evidence: Currently authors have only stated that the nucleocapsid antigen is what was detected, was any method to detect spike protein or RBD?

Reviewer #2: -Are the objectives of the study clearly articulated with a clear testable hypothesis stated? - NO

-Is the study design appropriate to address the stated objectives? - NO

-Is the population clearly described and appropriate for the hypothesis being tested? - No

-Is the sample size sufficient to ensure adequate power to address the hypothesis being tested? - NO

-Were correct statistical analysis used to support conclusions? - yes

-Are there concerns about ethical or regulatory requirements being met? - yes

**Results**

-Does the analysis presented match the analysis plan?

-Are the results clearly and completely presented?

-Are the figures (Tables, Images) of sufficient quality for clarity?

Reviewer #1: The results of the multivariable model do not match the goal of the analysis plan, recommend authors address whether they are trying to identify a causal model or identify predictors. 

Figure/Table placeholders are currently placed in the methods section, recommend authors consider putting these placeholders near the results they pertain to.

Reviewer #2: Does the analysis presented match the analysis plan? - yes

-Are the results clearly and completely presented? -yes

-Are the figures (Tables, Images) of sufficient quality for clarity? - too many

**Conclusions**

-Are the conclusions supported by the data presented?

-Are the limitations of analysis clearly described?

-Do the authors discuss how these data can be helpful to advance our understanding of the topic under study?

-Is public health relevance addressed?

Reviewer #1: Overall, nice discussion and public health relevance is addressed. However, I do a couple of concerns that authors should address in the limitations section:

-Authors do not address the time between potential exposure and seroconversion. We know that individuals can wane in detection overtime and this is a bias/flaw of the study design. This needs to be addressed of how this would impact the effect estimates; along if its differential/non-differential misclassification; and the degree to which the effect estimates may be impacted

Reviewer #2: Are the conclusions supported by the data presented? - NO

-Are the limitations of analysis clearly described? -yes 

-Do the authors discuss how these data can be helpful to advance our understanding of the topic under study? -Yes

-Is public health relevance addressed? - yes

**Editorial and Data Presentation Modifications?**

Reviewer #1: (No Response)

Reviewer #2: None

**Summary and General Comments**

Reviewer #1: (No Response)

Reviewer #2: Author’s efforts to conduct research work and publish this study entitled “SARS-CoV-2 infection and COVID-19 among 5 years-old Amazonian children and their association with poverty and food insecurity” are appreciable. However, there are many issues related to this study. Some of the comments are as follows.

1. selected study population from ongoing enrolled cohort cannot be equated with general population as the opportunities to interact with care provider, counselling and heath care seeking behaviour are likely to be different.

2. Inclusion of a control group from general population was desirable.

3. This data pertains to the pandemic survivors only. There is no information about those children who lost their life due to covid or related complications. 

4. Based on the reports from other countries, most vulnerable children for covid complications and adverse outcome are in the preschool age group (especially infants) or adolescence. Rationale for screening at 5 years of age is not clear.

5. Basic population characteristics with regard to immunisation status, anemia, chronic infections and other comorbidities are not addressed

6. Title of the study is not representative of work done and need modifications with regard to Covid sero-positivity

7. Conclusions are not supported by the study and represent more of general statement.

8. Author have listed many limitations of this study, substantiating a fact that it may not be possible to draw a meaningful conclusion from this study and possibly does not add much to existing knowledge.

PLOS authors have the option to publish the peer review history of their article (what does this mean?). If published, this will include your full peer review and any attached files.

Reviewer #1: No

Reviewer #2: No
---

## [Decision Letter · Decision Letter 1]

29 Apr 2022

Dear Dr. Cardoso,

Thank you very much for submitting your manuscript "SARS-CoV-2 seropositivity and COVID-19 among 5 years-old Amazonian children and their association with poverty and food insecurity" for consideration at PLOS Neglected Tropical Diseases. As with all papers reviewed by the journal, your manuscript was reviewed by members of the editorial board and by several independent reviewers. The reviewers appreciated the attention to an important topic. Based on the reviews, we are likely to accept this manuscript for publication, providing that you modify the manuscript according to the review recommendations. 

Due to the mixed reviews on the first evaluation, an additional reviewer was invited. The previous reviewer (1) noted that all issues were addressed, and it is in the editors' opinions that the issues raised by Reviewer 2 were adequately addressed. However, there are still some issues to be addressed, as specified by a new reviewer. As you have extensively revised the first submission, and the revision is now much more clear about the study design, analysis, and limitations, we believe the remaining issues may be easily resolved.

Sincerely,

Georgios Pappas

Associate Editor

Jeremy V. Camp, PhD

Deputy Editor

Please note editorial comments above when responding to the additional comments from reviewers.

Reviewer's Responses to Questions

**Key Review Criteria Required for Acceptance?**

**Methods**

-Are the objectives of the study clearly articulated with a clear testable hypothesis stated?

-Is the study design appropriate to address the stated objectives?

-Is the population clearly described and appropriate for the hypothesis being tested?

-Is the sample size sufficient to ensure adequate power to address the hypothesis being tested?

-Were correct statistical analysis used to support conclusions?

-Are there concerns about ethical or regulatory requirements being met?

Reviewer #1: (No Response)

Reviewer #3: 1. The retrospective study design is a limitation.

2. It is not clear how the study was planned and carried out. Is the design overall explorative? Specifically, was there any pre-plan of outcomes, or based on what were they selected? If the outcome selection was based on preliminary results, the quality (i.e. replicability) of the results will be low because of high risk of Type 1 error. In the same line of thought, were there any pre-planned statistical analysis plan? 

3. It is essential to address whether the findings are SARS-CoV-2/COVID19-specific, or in fact pertinent to airway infections in general: The analyses must include an (exploratory) analyses with another composite outcome of any cough, shortness of breath, loss of taste or smell (and no regard to +/- SARS-CoV-2 results). Such approach will test if (the same? or other?) socioeconomic factors determine symptoms often experienced by individuals having airway infections in general. Using symptoms of gastrointestinal infection as the explorative composite outcome, the same approach would test if (the same? or other?) socioeconomic factors determine infections in general. Such input is of outmost importance to the overall message: If socioeconomic factors determine any infection (which is likely), specific vaccination programs will not suffice when it comes to securing better health globally.

4. Based on what was predictors selected? Also after reading the authors' answer to prior reviewer comments, the processes of predictor selection and statistical modelling are not clearly described and argued for. For each round of analyses (crude, adjusted, final adjusted), please provide information on assumptions, criteria and evidence base, inclusive considerations about each factor (for example is household food insecurity considered a socioeconomic factor, or a somatic factor (hunger and malnutrition leading to less efficient organ function and immune responses)?.

5. Figure 2: Information on deaths in the overall population is irrelevant and may be misleading. Please delete information on mortality from the figure.

**Results**

-Does the analysis presented match the analysis plan?

-Are the results clearly and completely presented?

-Are the figures (Tables, Images) of sufficient quality for clarity?

Reviewer #1: (No Response)

Reviewer #3: 1. Figure 3: How can the percentages be the excact same in the two analyses?

2. Any considerations about multiplicity?

3. Line 265 - 267: Why "interestingly"? It is well known - also for SARS-CoV-2 - that crowding, especially indoor crowding, increases the risk of transmission. Please note, as mentioned above, that transmission patterns cannot be inferred from the present study where no information on index case is available (please see Discussion below).

**Conclusions**

-Are the conclusions supported by the data presented?

-Are the limitations of analysis clearly described?

-Do the authors discuss how these data can be helpful to advance our understanding of the topic under study?

-Is public health relevance addressed?

Reviewer #1: (No Response)

Reviewer #3: Lines 290 - 293 the authors conclude that their data point towards small children being a source of SARS-CoV-2-household transmission. That cannot be concluded based on the present study, since data on the timing of infection (i.e. index cases) were not available. It may be most likely that the SARS-CoV-2 is introduced to the family by the adolescents or adults who have more contacts outside the family.

**Editorial and Data Presentation Modifications?**

Reviewer #1: (No Response)

Reviewer #3: (No Response)

**Summary and General Comments**

Reviewer #1: Thank you for addressing all of my comments/feedback, the manuscript reads much clearer. 

I just have two very minor revisions: 

1. Throughout the text, some sentences appear in a lighter text and different font. Recommend revising. 

2. From my prior concern: "Authors do not address the time between potential exposure and seroconversion. We know

that individuals can wane in detection overtime and this is a bias/flaw of the study design. This

needs to be addressed of how this would impact the effect estimates; along if its

differential/non-differential misclassification; and the degree to which the effect estimates may

be impacted." 

It is recommended authors add a summarized version of their response/rationale to this comment in the limitations section of the manuscript. Ensure you state regardless of the method used the type of misclassification bias and the degree this may impact your effect estimates.

Reviewer #3: Thank you for letting med review this interesting and relevant study. 

Major revisions are needed to Methods, Results and Discussion.

PLOS authors have the option to publish the peer review history of their article (what does this mean?). If published, this will include your full peer review and any attached files.

Reviewer #1: No

Reviewer #3: Yes: Lone Graff Stensballe

Figure Files:

Data Requirements:

Reproducibility:

References

---

## [Decision Letter · Decision Letter 2]

30 May 2022

Dear Dr. Cardoso,

Thank you very much for submitting your manuscript "SARS-CoV-2 seropositivity and COVID-19 among 5 years-old Amazonian children and their association with poverty and food insecurity" for consideration at PLOS Neglected Tropical Diseases. As with all papers reviewed by the journal, your manuscript was reviewed by members of the editorial board and by several independent reviewers. The reviewers appreciated the attention to an important topic. 

The manuscript will be evaluated upon the next revision by the editorial staff and will not go to reviewers again. It can be considered accepted, provided that the minor issues raised by Reviewer 3 will be addressed.

Sincerely,

Georgios Pappas

Associate Editor

Jeremy Camp

Deputy Editor

The manuscript will be evaluated upon the next revision by the editorial staff and will not go to reviewers again. It can be considered accepted, provided that the minor issues raised by Reviewer 3 will be addressed.

Reviewer's Responses to Questions

**Key Review Criteria Required for Acceptance?**

**Methods**

-Are the objectives of the study clearly articulated with a clear testable hypothesis stated?

-Is the study design appropriate to address the stated objectives?

-Is the population clearly described and appropriate for the hypothesis being tested?

-Is the sample size sufficient to ensure adequate power to address the hypothesis being tested?

-Were correct statistical analysis used to support conclusions?

-Are there concerns about ethical or regulatory requirements being met?

Reviewer #1: (No Response)

Reviewer #3: OK

**Results**

-Does the analysis presented match the analysis plan?

-Are the results clearly and completely presented?

-Are the figures (Tables, Images) of sufficient quality for clarity?

Reviewer #1: (No Response)

Reviewer #3: Please note that the new Table S1 presenting predictors of symptoms suggestive of respiratory infection in the SARS-CoV-2 negative children show significant associations between the socioeconomic factors "household wealth index being beyond 1 (the poorest)", and "mothers schooling being beyond 9 years" and a significantly decreased risk of symptoms suggestive of respiratory infection. Thus, the concept socioeconomic status certainly predicts respiratory symptoms in the children under study, and this prediction is not SARS-CoV-2 specific. Although in the analysis of the subgroup of SARS-CoV-2 positive children, the associations were significant for other socioeconomical factors, namely "non-white mothers" and "food insecurity". 

And residual confounding is - of course - present in the present observational study and must be acknowledged.

**Conclusions**

-Are the conclusions supported by the data presented?

-Are the limitations of analysis clearly described?

-Do the authors discuss how these data can be helpful to advance our understanding of the topic under study?

-Is public health relevance addressed?

Reviewer #1: (No Response)

Reviewer #3: Significant associations were found between socioeconomic factors and respiratory symptoms, no matter if the children were SARS-CoV-2 positive or negative. This means that for example covid19-vacicination is not the cure. Improved socio-economy is a much better cure! I find that message of utmost importance. The point of socio-economy predicting transmission and severity of a broad range of infections, in the present study defined as respiratory symptoms, is indeed important and should be elaborated upon in the discussion.

**Editorial and Data Presentation Modifications?**

Reviewer #1: (No Response)

Reviewer #3: (No Response)

**Summary and General Comments**

Reviewer #1: Thank you for addressing the comment on misclassification bias. I have no further recommendations/revisions.

Reviewer #3: Thank you for letting me read this interesting and relevant manuscript again.

PLOS authors have the option to publish the peer review history of their article (what does this mean?). If published, this will include your full peer review and any attached files.

Reviewer #1: No

Reviewer #3: Yes: Lone Graff Stensballe

Figure Files:

Data Requirements:

Reproducibility:

References

---

## [Editor Report · Decision Letter 3]

13 Jun 2022

Dear Dr. Cardoso,

We are pleased to inform you that your manuscript 'SARS-CoV-2 seropositivity and COVID-19 among 5 years-old Amazonian children and their association with poverty and food insecurity' has been provisionally accepted for publication in PLOS Neglected Tropical Diseases.

Best regards,

Georgios Pappas

Associate Editor

Jeremy Camp

Deputy Editor

<style type="text/css">p.p1 {margin: 0.0px 0.0px 0.0px 0.0px; line-height: 16.0px; font: 14.0px Arial; color: #323333; -webkit-text-stroke: #323333}span.s1 {font-kerning: none

</style>

---

## [Editor Report · Acceptance letter]

1 Jul 2022

Dear Dr. Cardoso,

We are delighted to inform you that your manuscript, "SARS-CoV-2 seropositivity and COVID-19 among 5 years-old Amazonian children and their association with poverty and food insecurity," has been formally accepted for publication in PLOS Neglected Tropical Diseases.

Best regards,

Shaden Kamhawi

co-Editor-in-Chief

Paul Brindley

co-Editor-in-Chief
